# MSDU-Net: A Multi-Scale Dilated U-Net for Blur Detection

**DOI:** 10.3390/s21051873

**Published:** 2021-03-08

**Authors:** Xiao Xiao, Fan Yang, Amir Sadovnik

**Affiliations:** 1School of Telecommunications Engineering, Xidian University, Xi’an 710071, China; fyang_jfsl@stu.xidian.edu.cn; 2Department of Electrical Engineering & Computer Science, The University of Tennessee, Knoxville, TN 37996, USA; asadovnik@utk.edu

**Keywords:** blur detection, image segmentation, U-shaped network

## Abstract

A blur detection problem which aims to separate the blurred and clear regions of an image is widely used in many important computer vision tasks such object detection, semantic segmentation, and face recognition, attracting increasing attention from researchers and industry in recent years. To improve the quality of the image separation, many researchers have spent enormous efforts on extracting features from various scales of images. However, the matter of how to extract blur features and fuse these features synchronously is still a big challenge. In this paper, we regard blur detection as an image segmentation problem. Inspired by the success of the U-net architecture for image segmentation, we propose a multi-scale dilated convolutional neural network called MSDU-net. In this model, we design a group of multi-scale feature extractors with dilated convolutions to extract textual information at different scales at the same time. The U-shape architecture of the MSDU-net can fuse the different-scale texture features and generated semantic features to support the image segmentation task. We conduct extensive experiments on two classic public benchmark datasets and show that the MSDU-net outperforms other state-of-the-art blur detection approaches.

## 1. Introduction

Image blurring is one of the most common types of degradation caused by the relative motion between the sensor and the scene during image capturing. Object motion, camera shake, or objects being out of focus will cause the image to be blurred and reduce the visual quality of the image. This procedure can be regarded as the convolution of a clear image and a blur kernel, which is shown as:(1)B=I⊗K+N
where B is the blurred image, I is the clear image, K is the blur kernel, ⊗ is the convolution, and N is the noise. Since the blur kernel is usually unknown and varies greatly in size, weight, shape, and position, the estimation of blur kernel is an ill-posed inverse problem. The first important step for blur estimation is to detect the blurred regions in an image and separate them from clear regions. Image blurring can be categorized into two main types: defocus blur caused by defocusing and motion blur caused by camera or object motion. Blur detection plays a significant role in many potential applications, such as salient object detection [1,2], defocus magnification [3,4], image quality assessment [5,6], image deblurring [7], image refocusing [8,9], and blur reconstruction [10,11].

In the past few decades, a series of blur detection methods based on hand-crafted features have been proposed. These methods exploit various hand-crafted features related to the image gradient [12,13,14,15] and frequency [3,7,16,17]. They tend to measure the amount of feature information contained in different image regions to detect blurriness, as the blurred regions usually contain fewer details than the sharp ones. However, these hand-crafted features are usually not good at differentiating sharp regions from a complex background and cannot understand semantics to extract sharp regions from a similar background.

Recently, deep convolutional neural networks (DCNNs) have made vital contributions to various computer vision tasks, such as image classification [18,19], object detection [20,21] and tracking [22,23], image segmentation [24,25], image denoising [26,27], image interpolation [28], and super resolution [29,30]. Several DCNN-based methods have been proposed thus far to address blur detection [31,32,33,34,35,36]. Some methods [35,36] use patch-level DCNNs to learn blur features in every patch, and others [31,32,33,34] use a fully convolutional network trained at different scales to learn blur features from multiple scales. They use many different types of various extractors to capture the essential feature information to detect blur information. In this paper, from a different perspective we consider blur detection as an image segmentation problem. Hence, we can learn some successful methods and tricks from the image segmentation area. Inspired by some classical image segmentation approaches such as U-net [25], we propose a U-shape multi-scale dilated network called MSDU-net to detect blurred regions. In this model, the U-shape architecture uses skip connections to combine the shallow features and deep features smoothly. Moreover, this model can fully utilize of the texture and semantic features of an image. In this work, we find that texture information can be used to describe the degree of blur, and semantic information plays a vital role in measuring the blurriness of each region in an image. Therefore, we propose a group of multi-scale feature extractors to capture different-scale texture features in a synchronous manner. Additionally, We apply dilated convolution with various dilation rates and strides to capture the texture information with different receptive fields [37,38]. In particular, we use low-dilation convolution and small-stride convolution to capture the texture information on a small scale and use a high dilation rate convolution and a large stride convolution to capture texture information on a large scale.

To sum up, our main contributions are as follows:We proposed a group of extractors with dilated convolution to capture multi-scale texture information on purpose rather than using a fully convolutional network multiple times on the different scales of the image.We designed a new model with our extractors based on U-net, which can fuse the multi-scale texture and semantic features simultaneously to improve the accuracy.Most methods only can detect the defocus blur or the motion blur, but our method addressed the blur detection, ignoring the specific cause of the blur and thus could detect both defocus blur and motion blur. Compared with the state-of-the-art blur detection methods, the proposed model obtained F${}_{\sqrt{0.3}}$-measure scores of more than 95% in all the three datasets.

The rest of the paper is organized as follows: In Section 2, we introduce the traditional methods and deep learning methods and also some successful methods for image segmentation. In Section 3, we propose our model and describe the details of the neural network. In Section 4, we use our model with public blur detection datasets and compare our experimental results with those of other state-of-the-art methods. In Section 5, we conclude all the work of the paper.

## 2. Related Work

In this section, we will introduce the related work in the area of blur detection. We will show two main streams: (1) traditional views of blur detection and (2) regarding the blur detection problem as an image segmentation problem.

Previous methods of blur detection can be divided into two categories: methods based on traditional hand-crafted features and methods based on deep learning neural networks. In the first category, various hand-crafted features exploit gradient and frequency and can describe the information of regions. For example, Su et al. [39] used the gradient distribution pattern of the alpha channel and a metric based on singular value distributions together to detect the blurred region. In 2014, Shi et al. [7] made use of a series of gradient, Fourier domain, and data-driven local filter features to enhance the discriminative power for blur detection. In 2015, to enable feature extractors to distinguish noticeable blur reliably from unblurred structures, Shi et al. [16] improved feature extractors via sparse representation and image decomposition. Yi et al. [12] used a designed metric based on local binary patterns to detect the defocus regions. Tang et al. [17] designed a log-averaged spectrum residual metric to obtain a coarse blur map and iterate to a fine result based on the regional correlation. Golestaneh et al. [13] used a discrete cosine transform based on a high-frequency multi-scale fusion and sorted the transform of gradient magnitudes to detect a blurred region. In summary, traditional approaches aim to improve the accuracy with more meaningful and representative hand-crafted features, which are more interpretable. However, designing such features is difficult and the performance various among different datasets.

Because of the outstanding performance in high-level feature extraction and parameter learning, deep convolutional neural networks have reached a new state-of-the-art level in blur detection. Firstly, Park et al. [36] and Huang et al. [35] both used patch-level DCNNs in their methods to caputre local features more robustly to help detect blurred regions. Although patch-level DCNN methods use DCNNs, they do not make full use of the advantages of DCNNs. In 2018, Zhao et al. proposed a multi-stream bottom-top-bottom fully convolutional network [40], and Ma et al. also proposed an end-to-end fully convolution network [33]. Both the two methods are based on fully convolutional networks, and they novelly use high-level semantic information to help with blur detection. In order to increase the efficiency of the network, Tang et al. proposed a new blur detection deep neural network [34] by recurrently fusing and refining multi-scale features. Zhao et al. designed a cross-ensemble network [32] with two groups of defocus blur detectors, which were alternately optimized with cross-negative and self-negative correlation losses to enhance the diversity of features. With the application of DCNNs in computer vision, more solutions have been proposed for blur detection. Making the network deeper or wider to catch more useful features has been proven to be possible, but this kind of method is so dull that it incurs unnecessary resource consumption.

Except for the traditional view of the blur detection problem, blur detection problems can also be regarded as image segmentation problems. As we know, fully convolutional networks (FCNs) [41], which train end-to-end and pixel-to-pixel on semantic segmentation, exceed the previous best results without further machinery. Some classical architectures, such as DeepLab models [42,43,44] and U-net [25], have good performance in image segmentation. In DeepLab models [42,43,44], dilated convolution has been used to efficiently obtain feature maps with a larger receptive field.

The U-shape network was first proposed in [25] to address biomedical image segmentation, and it only had a few training samples. To make the best use of the limited samples, U-net [25] combines skip layers and learned deconvolution to fuse the different-level features of one image for a more precise result. Because of its outstanding performance in biomedical datasets with simple semantic information and a few fixed features, there are many further studies based on it, such as VNet [45], which is a U-shaped network that uses three-dimensional convolutions; UNet++ [46], which is a U-shaped network with more dense skip connections; Attention U-net [47], which combines U-shaped networks with an attention mechanism; ResUNet [48], which implements a U-shaped network with residual convolution blocks; TernausNet [49], which uses the pre-trained encoder to improve a U-shape network; MDU-Net [50], which densely connects the many scales of a U-shaped network; and LinkNet [51], which attempts to modify the original U-shaped network for efficiency.

The achievements of a U-shape network provide a number of valuable references for us to solve blur detection. In particular, the U-shape architecture can fuse different feature maps with different receptive fields. Thus, we designed our network on the basis of U-net [25].

## 3. Proposed MSD-Unet

Our model consists of two parts: a group of extractors and a U-shaped network. First, we used a group of extractors to capture multi-scale texture information from the images. Then, we inserted the extracted feature maps into each contracting step of the U-shaped network and integrated the extracted feature maps and the contracted feature maps together. Finally, we use a soft-max layer to map the feature matrix to the segmentation result. The whole model is shown in detail in Figure 1.

### 3.1. Basic Components

The U-shaped architecture can fuse the different-scale texture information to get a better result. Thus, we chose the U-shaped network to fuse the different-scale texture information. We used a group of extractors to capture the multi-scale texture information. Furthermore, in order to improve the efficiency of the extractors we used dilated convolution layers in extractors, which can enlarge the receptive field without increasing the parameters.

Dilated convolution, also called atrous convolution, was originally developed in algorithms for wavelet decomposition [52]. The main idea of the dilated convolution is to insert a hole between the pixels in the convolutional kernel to increase its receptive field. The receptive field is the size of the area mapped in the original image by the pixels on the feature map of each layer of the convolutional neural network, which is equivalent to how large the pixels in the high-level feature map are affected by the original image. The dilated convolution can effectively improve the extraction ability of convolution kernels for more features with a fixed number of parameters. If we set the center of the convolution kernel as the origin of the coordinates, for a 2D convolution kernel with size k×k, the result of the *r* dilation can be expressed as follows:(2)α=r−1
(3)$$S_{d}=S_{o}+\left(S_{o}-1\right)\cdot \alpha$$
where $S_{d}$ is the size of the dilated convolution kernels, $S_{o}$ is the size of the origin convolution kernel, and α is the dilation factor.
(4)$$K_{d}\left(x,y\right)=\left\{\begin{array}{cc} K_{o}\left(i,j\right) & \mathrm{if},\;x=i\cdot \alpha ,y=j\cdot \alpha \\ 0 & \mathrm{else} \end{array}\right.$$
where $K_{d}\left(x,y\right)$ is a single parameter in the dilated convolution kernel and $K_{o}\left(x,y\right)$ is a single parameter in the origin convolution kernel. In Figure 2, we can see a 3×3 convolutional kernel change to a dilated convolutional kernel with a 2 dilation. With the deep learning method, we can use a deeper network to catch the more abstract features. However, whether the region is blurred depends on the direct features. Thus, we need to increase the receptive field by expanding the size of the convolution kernel without making the network deeper. In our method, we exploited dilated convolutions to design a group of extractors which could extract texture information but needed no more additional parameters. In other words, with the same number of parameters the kernels can have a bigger receptive field, as is shown as Equation (Equation 5):(5)$$F\left(i,j-1\right)=\left(F\left(i,j\right)-1\right)\cdot stride+r\cdot \left(S_{d}-1\right)+1\;\;,i\geq j\geq 2$$

In Formula (Equation 5), F(i,j) is the local receptive field of the i-th layer to the j-th layer and stride is the kernel moving step. If $S_{d}$ and stride are fixed, F(i,j) increases with dilation *r*. Additionally, this recurrence formula has the initial condition:(6)F(1,1)=1

This means the pixels in the source image are only effected by themselves.

Skip connections combine the straight shallow features and abstract deep features, which can make the network notice shallow texture information and deep semantic information and thus gain a more precise result. As we know, the greater the number of convolution layers stacked, the greater the amount of high-level abstract information extracted. Traditional encoder-decoder architectures can extract high-level semantic information and perform well in panoramic segmentation that contains abundant high-level information. However, if we have to make images segment with the data only containing poor high-level information, such as cell splitting, MIR image segmentation, and satellite image segmentation, we should efficiently exploit the low-level information. The skip connections retain the low-level features in the shallow layers and combine them with the high-level features after deep layers, which can make the best use of both high-level and low-level information. The low-level information means the feature maps which have a small receptive field, and the high-level information means the feature maps which have a big receptive field. We can use the skip connections to fuse the feature maps with different receptive fields efficiently.

For our task, the low-level information of the gradient and the frequency can describe the absolute degree of blur, and the high-level information of global semantics can help to judge whether the regions are blurred. As a result, the skip connections can make our model robust to various backgrounds.

### 3.2. Model Details

It can be seen that U-net has two paths: the contracting path and the expansive path. However, in order to combine with the multi-scale texture extractors, we modified the contracting path of the U-net to receive different-scale texture feature matrices at every stage. In this section, we describe the detail of the extractors and the U-shape network in our model.

We designed the extractors, aiming at capturing the multi-scale texture feature. Firstly, the source image is fed into the dilated convolution layers. The dilation rates of this layer in different extractors are 1,2,2,2 correspondingly. All the kernel sizes in this layer are 3×3. Secondly, the outputs of the dilated convolution layers are sent to normal convolution layers with a ReLU activation function and batch normalization layers. Then, we used the max pooling layers with strides of 1,2,4,8 and the kernel sizes of 2×2, 2×2, 4×4, and 8×8 to shrink the sizes of the feature maps. This makes the output feature maps of extractor the same as the size of the feature map of each contracting path in U-shaped architectures. After that, all the output feature map of the extractors can be contacted with the corresponding feature maps of each contracting path in U-shaped architectures.

The contracting path which receives the outputs of texture extractors and integrates them through concatenation, convolution, and pooling decreases the length and width of the feature matrices and increases the channel dimensions. The expansive path uses transposed convolutions to restore the resolution of feature matrices and concatenates them with the feature matrix that has the same size in the contracting path through skip connections. The U-shaped architecture uses skip layers to concatenate the feature channels of the two paths in the upsampling part, which allows the network to propagate semantic information to higher-resolution layers that contain local texture information.

Because the contracting path and the expansive path are almost symmetric, the whole architecture is vividly called U-shaped architecture. The blocks in the contracting path follow the typical architecture of a U-net [25], which stacks two 3×3 convolution layers that followed by a ReLU and a 2×2 max pooling layers with stride 2. The input feature maps of every step in the contracting path are combined with the output of the last step and the corresponding extractor. The expansive path is almost the same as that of the U-net. It is consists of a 2×2 transposed convolution that halved the number of feature channels, and two 3×3 convolutions, each followed by a ReLU. The input feature maps of every step in the expansive path are combined with the output of the last step and the corresponding output maps of the contracting path.

## 4. Experiments

We compared MSDU-net with other methods on the public datasets and analyzed the results on different indicators. We also conducted experiments to prove the effect of the components in MSDU-net. We resize all the pictures into 256 × 256 to prevent from causing insufficient memory. We use the dice coefficient as the loss function, which is shown as follows:(7)$$Dice=\frac{\left|P\cap G\right|}{\left|P|\cup |G\right|}$$
where the P is the blur pixel set we detected, and the G is the blur pixel set from the ground truth. This loss function is also called IoU, Intersection over Union.

### 4.1. Datasets and Implementation

We performed our experiments on two publicly available benchmark datasets for blur detection. CUHK [7] is a classical blur detection dataset in which 296 images are partially motion-blurred and 704 images are defocus-blurred. DUT [40] is a new defocus blur detection dataset that consists of 500 images as the test set and 600 images as the training set. We separated the CUHK blur dataset into a training set, which included 800 images, and a test set, which included 200 images that had the same ratio of motion-blurred images and defocus-blurred images. As the number of training samples was limited, we enlarged the training set by horizontal reversal at each orientation. Because that some state-of-the-art methods were designed solely for defocus blur detection, when we compared with these methods on the CUHK blur dataset we only used the 704 defocus-blurred images from CUHK. We separated them into a training set, which included 604 images, and a test set, which included 100 images. Our experiments were performed on these three datasets (CUHK, DUT, and CUHK-defocus).

We implemented our model in Pytorch and trained our model on a machine equipped with an Nvidia Tesla M40 GPU with 12 GB. We optimised the network by using the stochastic gradient descent (SGD) algorithm with a momentum of 0.9, a weight decay of $5e^{-4}$ and a learning rate of 0.01 in the beginning and reduced by a factor of 0.1 every 25 epochs. We trained with a batch size of 16 and resized the input images to 256×256, which required 10 GB of GPU memory for training. We used our enhanced training set of 5200 images to train our model for a total of 100 epochs.

### 4.2. Evaluation Criteria and Comparison

We varied the threshold to produce a segmentation of sharpness maps to draw the precison and recall curve.
(8)$$precision=\frac{R\cap R_{g}}{R}\;\;,\;recall\;=\frac{R\cap R_{g}}{R_{g}}$$
where R is the set of pixels in the segmented blurred region and $R_{g}$ is the set of pixels in the ground truth blurred. The threshold $T_{seg}$ value is sampled at every integer within the interval [0,255].

The F-measure, which is an overall performance measurement, is defined as Equation (Equation 9):(9)$$F_{\beta }=\frac{\left(1+\beta^{2}\right)\cdot precision\cdot recall}{\beta^{2}\cdot precision+recall}$$
where β is the weighting parameter ranging from 0 to 1. In our study, $\beta^{2}=0.3$, as in [12], is used to emphasize the precision. Precision means the percentage of blur pixels being correctly detected, and recall is the fraction of detected blur pixels in relation to the ground truth number of blur pixels. A larger *F* value means a better result.

Mean absolute error (MAE) can provide a good measure of the dissimilarity between the ground truth and the blurred map.
(10)$$\mathrm{MAE}=\frac{1}{W\cdot H}\sum_{x=1}^{W}\sum_{y\;=1}^{H}\left|G\left(x,y\right)-M_{final}\left(x,y\right)\right|$$
where x,y stand for pixel coordinates. G is the ground truth map and $M_{final}$ is the detected blur region map. *W* and *H* stand for the width and the height of the $M_{final}$ (or G), respectively. A smaller MAE value usually means that $M_{final}$ is closer to G.

We compared our method against nine other state-of-the-art methods, including deep learning-based methods and hand-crafted features methods: DeF [34], CENet [32], BTBNet [40], DBM [33], HIFST [13], SS [17], LBP [12], JNB [16], and DBDF [7]. In Figure 3, we showed some defocus-blurred cases of the visual comparison results. These cases include various scenes with cluttered backgrounds or similar backgrounds and contain complex boundaries of objects, which make it difficult to separate the sharp regions from the images. In Figure 4, we show some motion-blurred cases of the visual comparison result of different methods.

We also drew accurate precision-recall curves and F-measure curves to study the capabilities of these methods through statistical calculation. Figure 5 shows that our improvement progress on all the three tests, and particularly on the CUHK dataset which contains both defocus-blurred images and motion-blurred images. Our method boosts the precision within the entire recall range, where the improvement could be as large as 0.2. Furthermore, in Figure 6 the F-measure curves of our methods are all over 0.9, which are the best on each dataset. Table 1 shows that our method consistently performs favourably against other methods on the three data sets, which indicates the superiority of our method over the other approaches.

### 4.3. Ablation Analysis

Although U-shaped networks with skip layers have been applied in BTBNet, we performed supplementary experiments to verify the significance of the skip connections. To control the variables, we built a new model that is similar to our original model except that there are no skip layers, by using the CUHK blur dataset for training. By the comparison of the result, we found that the model without skip connections could not precisely segment the edges of objects in Figure 7. As a result, the model skip connections have a lower F${}_{\sqrt{0.3}}$-measure score and a higher MEA score, as in Table 2.

Multi-scale extractors with dilated convolution aim to extract multi-scale texture features to improve the precision of the blurred map. To verify its effect, we compared our network with the classical U-net which does not have multi-scale extractors. Figure 8 shows that the results of the U-net [25] without the multi-scale extractors are disturbed by backgrounds of shallow depths. Because of the multi-scale extractors, our model was so sensitive to the degree of blur that it could accurately separate the blur region. As a result, our model had a higher F${}_{\sqrt{0.3}}$-measure score and a lower MEA score in Table 3. Further, we replaced 3×3 dilated convolution kernels with the 5×5 normal convolution kernels, which had the same receptive field. However, as shown in Table 3, our model performed slightly worse than the model using 5×5 normal convolution kernels. However, our model save millions of parameters by using dilated convolutions.

## 5. Conclusions

In this work, we regarded blur detection as an image segmentation problem. We designed a group of multi-scale extractors with dilated convolutions to capture the different scale texture information of blur images. Then, we combined the extractors with the U-shaped network to fuse the shallow texture information and the deep semantic information. Taking advantage of the multi-scale texture information and the semantic information, our method performed better on the scenes with cluttered backgrounds or similar backgrounds and objects which contained complex boundaries. We tested our model on three datasets. The experimental results on three datasets proved that our method outperforms state-of-the-art methods in blur detection. Furthermore, our work could be applied to foreground and background segmentation, image quality evaluation, and so on. In the future, we will improve our model to not only detect the blur region but also to distinguish the degree of blurring of different regions and make our model robust and adapt to data outside the datasets.

## Figures and Tables

**Figure 1 sensors-21-01873-f001:**
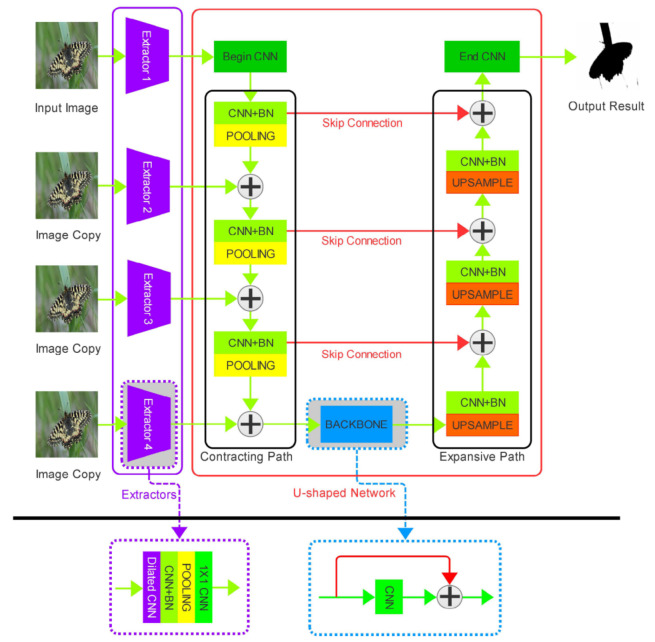
Detailed diagram of our model. Our model can be divided into two parts: a group of feature extractors are shown in the purple box and the U-shaped network is shown in the red box. In the U-shaped architecture, it can be divided into the contracting path, the expansive path, and the backbone. All the “+” signs in the figure mean connecting the two parts of feature in the channel dimension. Different colors of the blocks mean that the blocks have different functions: green blocks mean convolution; yellow blocks mean pooling; orange blocks mean upsampling; light green blocks mean convolution and batch normalization; purple blocks mean dilated convolution.

**Figure 2 sensors-21-01873-f002:**
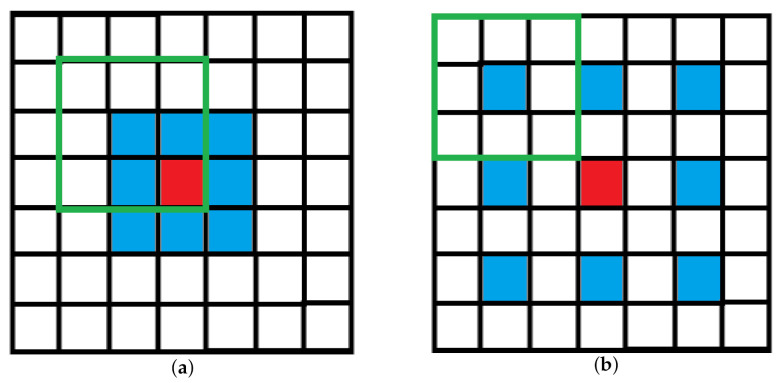
Dilated convolutional kernel has a bigger receptive field than normal convolutional kernel. (**a**) Normal 3×3 convolutional kernel; (**b**) dilated 3×3 convolutional kernel.

**Figure 3 sensors-21-01873-f003:**
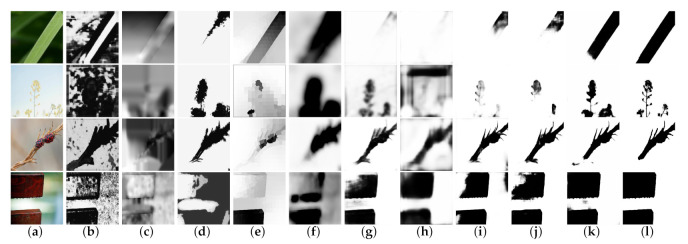
Defocus blur maps of generated using different methods. In the visual comparison, we can find that our method performed better in the scenes with a similar background or cluttered background. (**a**) Input, (**b**) DBDF14 [7], (**c**) JNB15 [16], (**d**) LBP16 [12], (**e**) SS16 SS [17], (**f**) HiFST17 [13], (**g**) BTB18 [40], (**h**) DBM18 [33], (**i**) DeF19 [34], (**j**) CENet19 [32], (**k**) MSDU-net, (**l**) GT(Ground Truth).

**Figure 4 sensors-21-01873-f004:**
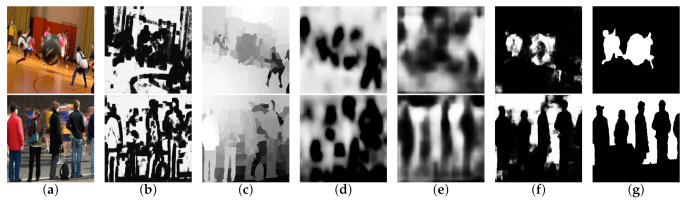
Motion blur maps generated using different methods. In the visual comparison, our method performed better than the other methods. (**a**) Input, (**b**) DBDF14 [7], (**c**) SS16 [17], (**d**) HiFST17 [13], (**e**) DBM18 [33], (**f**) MSDU-net, (**g**) GT.

**Figure 5 sensors-21-01873-f005:**
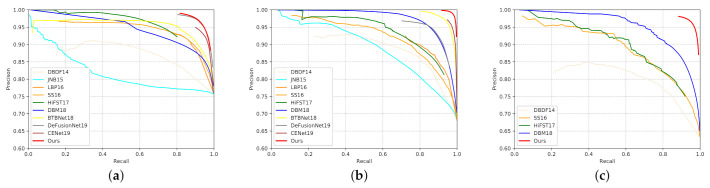
Comparison of the precision-recall curves of different methods on three test sets. The curves of our method are more than 95%. In particular, our method achieved an improvement of more than 0.2 progress in precision than the other method on the CUHK. (**a**) DUT test set; (**b**) CUHK${}^{*}$ test set(without motion blurred images); (**c**) CUHK test set

**Figure 6 sensors-21-01873-f006:**
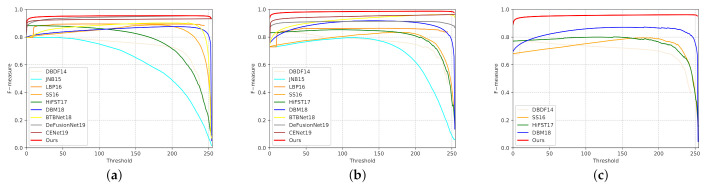
Comparison of F${}_{\sqrt{0.3}}$-measure curves of different methods on three test sets. The curves of our method are the highest curves on the three test sets. (**a**) DUT test set; (**b**) CUHK${}^{*}$ test set(without motion blurred images); (**c**) CUHK test set

**Figure 7 sensors-21-01873-f007:**
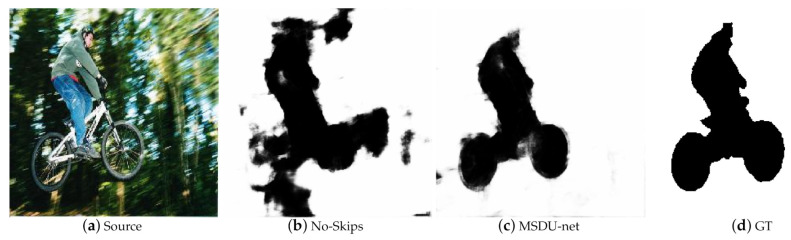
Visual comparison results between MSDU-net and the model without skip connections.

**Figure 8 sensors-21-01873-f008:**
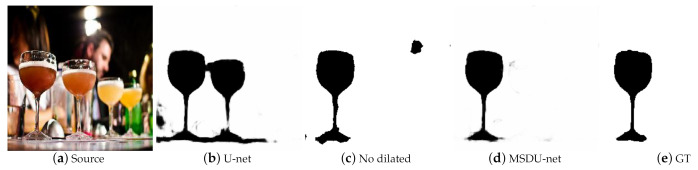
Visual comparison results among no dilatited (using 5×5 normal convolution kernels), our model and U-net.

**Table 1 sensors-21-01873-t001:** Quantitative comparison of F${}_{\sqrt{0.3}}$-measure and MEA scores. The best results are marked in bold. CUHK* in the table is the CUHK dataset excluding motion-blurred images, and “-” means that the methods are not designed for the motion blur. We compared these methods: DeF [34], CENet [32], BTBNet [40], DBM [33], HIFST [13], SS [17], LBP [12], JNB [16], and DBDF [7].

Datasets	Metric	DBDF	JNB	LBP	SS	HiFST	DBM	BTB	DeF	CENet	MSDU-Net
DUT	F${}_{\sqrt{0.3}}$	0.827	0.798	0.895	0.889	0.883	0.876	0.902	0.953	0.932	0.954
	MEA	0.244	0.244	0.168	0.163	0.203	0.165	0.145	0.078	0.098	0.075
CUHK*	F${}_{\sqrt{0.3}}$	0.841	0.796	0.864	0.834	0.853	0.918	0.963	0.914	0.965	0.976
	MEA	0.208	0.260	0.174	0.215	0.179	0.114	0.057	0.103	0.049	0.032
CUHK	F${}_{\sqrt{0.3}}$	0.768	-	-	0.795	0.799	0.871	-	-	-	0.953
	MEA	0.257	-	-	0.248	0.207	0.123	-	-	-	0.042

**Table 2 sensors-21-01873-t002:** Quantitative comparison of F${}_{\sqrt{0.3}}$-measure and MEA scores between our model and the model without skip connections.

Network	No Skip	MSDU-Net
F${}_{\sqrt{0.3}}$-measure	0.851	0.952
MEA	0.137	0.042

**Table 3 sensors-21-01873-t003:** Quantitative comparison of F${}_{\sqrt{0.3}}$-measure and MEA scores among no dilatited (using 5×5 normal convolution kernels), MSDU-net and U-net.

Network	U-Net	No Dilated (5×5)	MSDU-Net
F${}_{\sqrt{0.3}}$-measure	0.843	0.956	0.950
MEA	0.146	0.044	0.046

## Data Availability

All the datasets used in this work are the public datasets published on the Internet, which can be found from links: CUHK, DUT.

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
