# Peer review of "MSDU-Net: A Multi-Scale Dilated U-Net for Blur Detection"

_sensors, 2021, doi:10.3390/s21051873_

Round 1

Reviewer 1 Report

The results are good on the datasets trained on

The methodology and neural net proposed is straight forward and not really new 

maybe a little too much explanation of atrous convolution as it is a known technique

Reviewer 2 Report

1、The network MSDU-net proposed in this paper has obvious defects in the network structure: As can be seen from Figure 1, the input image and its copy are extracted by feature extractor and then merged with the layers of the left half of Unet. However, In the left half of Unet, there is a pooling layer between each layer, which is equivalent to 1 downsampling operation for each layer . Combined with the description of extractors in Figures 1 and 3.2, each extractor has only one pooling layer, and there is no relationship between different extractors. Except for the first image copy, how do the other image copies merge with the corresponding U-net layer after the extractor? Just imagine, can two images of different sizes be directly merged?

2、Even without considering the structure of MSDU-net, the network is not very innovative. You must know that the multi-scale input U-net is a relatively common network. MSDU-net just adds a dilated convolution layer to each multi-scale input. Moreover, from the experimental results (as shown in Table 3), this improvement does not seem to be obvious to the performance improvement. On the whole, the source of the advantages of MSDU-net over U-net is more like multi-scale input rather than dilated convolution.

3、In terms of details, this article has many errors, such as:
Line 74, "In the next year", which year?
Line 88, "Then", is there an inherent logical relationship between the two studies?
Line 109, "In [?]", what does it mean? Is the writing wrong?
Line 228, "resized the input images to 256", should be 256×256.

4、other suggestion:
(1) It is recommended to narrow the application range of MSDU-net or clarify the application environment of MSDU-net, because there are many reasons for blurring, and the principles of different types of blurring sources are different. It is difficult for a single network model to solve all problems with limited training samples. Even if it performs better on a certain data set, it may not work with another data set. For example, the CUHK blur dataset used in this article is a defocus-blurred image dataset.
(2) This article has repeatedly said "blur detection problems can also be regarded as an image segmentation problem", such as line 36 and line 99. For this conclusion, is there any more authoritative related research before? If not, this article should elaborate on this point in depth. For example, if the blur is caused by random noise, is it like image segmentation? You know, the objects in image segmentation have clear meanings.
(3) Line 137, "we used dilated convolution to replace standard convolution", is it replacing all convolutions? Or only use dilated convolution in extractor? If it is the former, it is best to modify Figure 1, especially adding dilated convolution to the U-net part; if it is the latter, then it must be clear here, which causes misunderstanding.
(4) For each comparison algorithm in section 4.2, such as DBDF14, JNB15, LBP16, the source should be clearly specified, and at least the corresponding paper should be indicated.
(5) 102-112, the content of the discussion is the application of FCN, U-net and other networks, which has nothing to do with the subject of this article. Is it necessary to discuss too much?

Reviewer 3 Report

This article is of interest and the results are convincing. The main novelty is the introduction of dilated convolution and the way to insert them in a Unet. In fact one can guess that equivalent results might be obtained by a usual Unet with filters of different sizes but at the price of adding many more parameters.

I do not understand why the authors are using the term "blur" as all they show in their experiments is that they are able to distinguish a shape, an object from the background, mainly when the latter is highly textured. The fact that they speak of "semantic"information enforces this point. It is not clear how the blur is defined. In fact as they are using a supervised model, it seems that what they call blur is essentially what has been labelled as such by an expert. In that sense the present work cannot be generalised to any type of noisy images

Also the degree  of blur is never discussed.

More details could be added with profit on the model : definition of the loss fonction, the ponderation of the weights, the output size of the Unet etc…

Round 2

Reviewer 2 Report

No more questions

Reviewer 3 Report

The authors have answered successfully the points that I stressed in my review, therefore I recommend to publish the paper.